# Developing a Novel Murine Meningococcal Meningitis Model Using a Capsule-Null Bacterial Strain

**DOI:** 10.3390/diagnostics14111116

**Published:** 2024-05-28

**Authors:** Viorela-I. Caracoti, Costin-Ș. Caracoti, Diana L. Ancuța, Fabiola Ioniță, Andrei-A. Muntean, Mangesh Bhide, Gabriela L. Popa, Mircea I. Popa, Cristin Coman

**Affiliations:** 1Faculty of Medicine, Microbiology Discipline II, Carol Davila University of Medicine and Pharmacy, 020021 Bucharest, Romania; viorela-ioana.nicolae@drd.umfcd.ro (V.-I.C.); costin-stefan.caracoti@drd.umfcd.ro (C.-Ș.C.); alexandru.muntean@umfcd.ro (A.-A.M.); gabriela.popa@umfcd.ro (G.L.P.); 2Cantacuzino National Military Medical Institute for Research and Development, Preclinical Testing Unit, 050096 Bucharest, Romania; ancuta.diana@cantacuzino.ro (D.L.A.); ionita.fabiola@cantacuzino.ro (F.I.); coman.cristin@cantacuzino.ro (C.C.); 3Faculty of Veterinary Medicine, University of Agronomic Sciences and Veterinary Medicine, 050097 Bucharest, Romania; 4Faculty of Veterinary Medicine, University of Veterinary Medicine and Pharmacy in Kosice, Komenskeho 73, 04181 Kosice, Slovakia; mangesh.bhide@uvlf.sk; 5Institute of Neuroimmunology of Slovak Academy of Sciences, Dubravska Cesta 9, 84510 Bratislava, Slovakia; 6Fundeni Clinical Institute Translational Medicine Centre of Excellence, 022328 Bucharest, Romania

**Keywords:** meningococcal meningitis, murine model, *Neisseria meningitidis*, capsule deficient meningococcus, capsule null locus

## Abstract

Background: *Neisseria meningitidis* (meningococcus) is a Gram-negative bacterium that colonises the nasopharynx of about 10% of the healthy human population. Under certain conditions, it spreads into the body, causing infections with high morbidity and mortality rates. Although the capsule is the key virulence factor, unencapsulated strains have proved to possess significant clinical implications as well. Meningococcal meningitis is a primarily human infection, with limited animal models that are dependent on a variety of parameters such as bacterial virulence and mouse strain. In this study, we aimed to develop a murine *Neisseria meningitidis* meningitis model to be used in the study of various antimicrobial compounds. Method: We used a capsule-deficient *Neisseria meningitidis* strain that was thoroughly analysed through various methods. The bacterial strain was incubated for 48 h in brain–heart infusion (BHI) broth before being concentrated and injected intracisternally to bypass the blood–brain barrier in CD-1 mice. This prolonged incubation time was a key factor in increasing the virulence of the bacterial strain. A total of three more differently prepared inoculums were tested to further solidify the importance of the protocol (a 24-h incubated inoculum, a diluted inoculum, and an inactivated inoculum). Antibiotic treatment groups were also established. The clinical parameters and number of deaths were recorded over a period of 5 days, and comatose mice with no chance of recovery were euthanised. Results: The bacterial strain was confirmed to have no capsule but was found to harbour a total of 56 genes coding virulence factors, and its antibiotic susceptibility was established. Meningitis was confirmed through positive tissue culture and histological evaluation, where specific lesions were observed, such as perivascular sheaths with inflammatory infiltrate. In the treatment groups, survival rates were significantly higher (up to 81.25% in one of the treatment groups compared to 18.75% in the control group). Conclusion: We managed to successfully develop a cost-efficient murine (using simple CD-1 mice instead of expensive transgenic mice) meningococcal meningitis model using an unencapsulated strain with a novel method of preparation.

## 1. Introduction

*Neisseria meningitidis* is a human-specific Gram-negative bacterium that is frequently responsible for severe cases of bacterial meningitis and is one of the few species that can cause large epidemics of such a disease [1]. It relies on several virulence factors, the capsule being one of the most important, as it prevents host phagocytosis and assists in the evasion of the host immune response [2]. Based on the capsular polysaccharide, multiple serogroups have been described as having A, B, C, Y, and W-135 major epidemiological importance [3]. However, unencapsulated strains exist, and they have been documented to have significant clinical implications, being capable of causing meningitis, bacteraemia, and other kinds of infections in humans worldwide [4].

Meningococcal vaccination may protect against the most common serogroups that cause meningococcal disease; however, the antibodies against A, C, Y, and W-135 *Neisseria* strains do not protect against unencapsulated strains because, in these cases, the vaccinal component is a group-specific polysaccharide [5,6]. Serogroup B meningococcal vaccines target other bacterial components, such as factor H-binding protein. Although these components may be present in other *N. meningitidis* strains, studies on protection against unencapsulated meningococci provided by serogroup-B vaccines are scarce to date [7].

Despite vaccine availability, vaccination is not universal, and based on the latest reports, *Neisseria meningitidis* remains one of the four main causes of bacterial meningitis worldwide [8]. It is a rapidly progressing infection that requires immediate diagnosis and efficient antibiotic treatment [9,10]. However, even when adequate care is provided, up to 15% of cases are still fatal [11]. Therefore, there is a great interest in the development of specific fast-acting anti-neisserial treatment molecules that can cross the blood–brain barrier. However, all new medications aimed at treating confirmed cases of meningococcal meningitis have to go through rigorous in vivo testing phases, which require animal infection models.

After going through the literature in our review [12], we have concluded that the existing meningococcal meningitis mouse models are scarce and rely on different factors, such as transgenic mice and hypervirulent encapsulated strains of *Neisseria meningitidis*. The most promising meningococcal meningitis animal model we found [13] was developed on the CD-1 mouse using a hypervirulent serotype C *Neisseria meningitidis* and a capsule-deficient variant of the same bacterial strain from Novartis.

Our aim was to develop an animal model using a widely available mouse strain and a clinically isolated capsule-deficient ST-823 strain of *Neisseria meningitidis*. The model’s veracity is strengthened by incorporating two antibiotic treatment groups meant to save the inoculated animals, by performing bacterial counts in sampled target organs, and by having all the necessary control groups to eliminate other possible causes of death. However, certain variables must be considered, including the broth inoculum incubation duration, which has proven to be a critical and novel feature of the model.

## 2. Materials and Methods

### 2.1. Animals

For this study, we have used 88 females aged 8–10 weeks, specific pathogen-free (SPF) CD-1 strain mice. Animals were provided by the Cantacuzino National Military Medical Institute for Research and Development (CI), Băneasa Animal Facility (BAF), and maintained in individually ventilated cages (Tecniplast, Buguggiate, Italy) with autoclaved bedding. Mice were allocated to groups with a maximum of six animals per cage, maintained under a 12 h/12 h (day/night) light cycle in a ventilated room with a controlled temperature of 20–24 °C and unrestricted access to food and filtered water. The CI experimental medicine and translational research platform where the animal experiments took place has an Environmental Enrichment Program. The humane endpoints were determined using a clinical examination based on the ARRIVE criteria [14], with clinical symptoms ranked according to severity. A veterinarian assessed the animal’s health. The animals were left to settle in the new environment for 1 week before conducting the experiments.

Ethics Statement: The animal experiment was approved by the Ethics Committee of the Cantacuzino National Military Medical Institute for Research and Development, Bucharest, and authorised by the Romanian Competent Authority. It was conducted in accordance with the national legislation 43/2014 and the EU Directive 63/2010 on the care, use, and protection of animals used for scientific purposes.

### 2.2. Bacterial Strain

We used a clinical strain of *Neisseria meningitidis* isolated from the cerebrospinal fluid of a hospitalised patient. This strain was later confirmed through whole-genome sequencing analysis to belong to an ST-823 strain from the ST-198 complex.

The bacterial strain was thoroughly characterised before the development of the animal model [14,15,16,17].

Agglutination tests for serogroup identification were performed using the Pastorex kit [18] for bacterial meningitis (Bio-Rad Laboratories, Inc., Hercules, CA, USA).

Antibiotic susceptibility testing was performed using the EUCAST guidelines for the standardised disc-diffusion antibiogram [19].

Whole-genome sequencing was performed on the strain using Illumina sequencing (Novaseq 6000). Annotation of the bacterial genome was performed using the Prokka annotation tool [20] embedded in Proksee [21] for the list of identified genes. Genes coding for virulence factors were identified by similarity searches across the hierarchical pre-build datasets performed using the VFanalyzer tool from the Virulence Factors of Pathogenic Bacteria Data Base (VFDB) [22]. Serotyping was also performed using the data from a Multi-Locus Sequence Typing (MLST) analysis [23] from the Centre for Genomic Epidemiology (CGE) based on the 7 gene signatures [24,25].

### 2.3. Bacterial Culture and 48-h Inoculum Preparation

Cryotube stocks of the meningococcus stored at −80 °C were thawed, and one 10 μL microbiological loop was inoculated on chocolate agar plates. The plates were then incubated at 36 ± 1 °C in an atmosphere containing 5–10% CO_2_ in order to obtain a pure culture. From the resulting 24-h culture, one isolated colony was inoculated on another chocolate agar plate and incubated under the same conditions.

From the resulting culture, a full 10 µL loop of bacterial growth was inoculated in 400 mL of brain–heart infusion (BHI) broth media supplemented with 5 mg/L iron dextran and incubated in aerobic conditions at 36 ± 1 °C for 24 h in the initial stages of our model’s development and for 48 h in the second stage.

After 24 h of incubation, 3 samples of the liquid culture were inoculated on chocolate agar plates and incubated for 24 h at 36 ± 1 °C in an atmosphere containing 5–10% CO_2_ to verify the purity of our culture. The broth was introduced back into the incubator for another 24 h. The purity of the broth culture was confirmed after 24 h by observing the growth aspect of the colonies and was certified by identification using the bacterial identification application of the matrix-assisted laser-desorption/ionisation time of flight (MALDI-TOF) mass spectrometry assay performed on a Bruker AutoFlex Speed using the MBT Compass (HT) IVD software (version 2023).

After another 24 h of incubation, the BHI broth culture was centrifuged at 2602× *g* for 10 min at room temperature.

The bacterial pellet obtained was transferred into an Eppendorf tube and resuspended in a small amount of BHI broth media with a 5 mg/L iron dextran supplement. Using a Tecan Sunrise Microplate Spectrophotometer, we adjusted the bacterial concentrate to an optical density of 3 measured at a wavelength of 630 nm. We have used the BHI broth supplemented with iron dextran as the baseline reading. This value was measured using the colony-forming unit (CFU) plate count method to correspond to an average of 1.5 × 10^10^ CFU/mL. To accomplish this, two 10^−6^ and 10^−7^ dilutions of the concentrate were obtained, and 10 μL from each dilution were inoculated on 3 chocolate agar plates and spread evenly with a sterile loop. The plates were incubated for 24 h at 36 ± 1 °C in an atmosphere containing 5–10% CO_2_. The resulting colonies were counted, and a mean bacterial concentration was calculated.

To obtain the inoculum used in the experiment, the bacterial concentrate was diluted to 1.5 × 10^9^ CFU/mL in BHI broth media. Considering the inoculum volume was 10 μL, this would correspond to 1.5 × 10^7^ CFU/mouse. Before intracisternal delivery, the concentration of the bacterial inoculum was verified one last time using the CFU plate count method in triplicate.

### 2.4. Inoculation Procedure

Before inoculation, all the animals were weighed, and their corporal temperature was measured using an intrarectal thermometer.

Two hours before intracisternal bacterial inoculation, the mice received a single intraperitoneal dose of 250 mg/kg of a pharmaceutical iron dextran supplement for veterinary use (FierDextran 20% from Farmavet Group).

The mice were anaesthetised intraperitoneally, according to individual weight, with a mixture of ketamine (50 mg/kg) and xylazine (3 mg/kg).

An ophthalmic lubricant was used to prevent eye drying.

Their cervical fur was shaved, and the exposed skin was cleaned with an antiseptic containing a 70% ethanol solution and 3% betadine.

The depth of anaesthesia was assessed by observing their reaction to touch and to the pain response when pinched.

The inoculation procedure was performed in a laminar airflow work cabinet.

The animal’s head was held with the left hand in a ventral flexed position. The craniocervical junction, which appears as a diamond-shaped indentation, is the inoculation area for the *cisterna magna* administration site. After being gently homogenised, the bacterial inoculum was delivered using a 30 G needle syringe.

The total volume of inoculum was 10 μL, corresponding to approximately 1.5 × 10^7^ CFU/mouse.

Following anaesthesia and inoculation, the mice were kept warm for an hour post-anaesthesia and then placed in cages until fully awakened.

### 2.5. Inactive Inoculum Control

An inactive bacterial inoculum control group was established to rule out any deaths induced by possible endotoxic shock or organic lesions caused by the inoculation procedure.

The inoculum was prepared as described above and inactivated with wet heat for one hour at 100 °C. To confirm the inactivation, two samples of inoculum were plated on chocolate agar and incubated for up to 48 h at 36 ± 1 °C in an atmosphere containing 5–10% CO_2_, and no bacterial growth was observed.

### 2.6. Diluted Inoculum Group

A 10^−2^ dilution of the bacterial concentrate was obtained and used, corresponding to 1.5 × 10^6^ CFU/mouse.

### 2.7. 24-h Incubated Inoculum

A neisserial broth culture was obtained using the same protocol as stated in [2,3], which was allowed to grow for only 24 h before being prepared for inoculation. The purpose of this inoculum was to compare it to the 48-h inoculum that proved successful in our study.

### 2.8. Antibiotic Treatment

Two different antibiotics were used based on the strain’s susceptibility and clinical indication.

We have agreed upon ceftriaxone as the drug of choice for the treatment of most bacterial meningitis, including meningococcal meningitis [26,27]. The animals received an initial intravenous (IV) dose of 50 mg/kg ceftriaxone (corresponding to an average volume of 10 µL/mouse, depending on the animal weight) 3 h after the intracisternal inoculation. Antibiotic treatment was administered every 12 h for the next 5 days (a total of 10 additional doses).

To further solidify the usefulness of the model, a second treatment group using ciprofloxacin chlorhydrate was established. This antibiotic was chosen since it is used in the prophylaxis of meningococcal meningitis after a possible exposure event. The animals received an initial IV dose of 25 mg/kg ciprofloxacin (corresponding to an average volume of 200 µL/mouse, depending on the animal weight) 3 h after the bacterial inoculation and thereafter a single identical dose daily for the next 5 days.

The control groups received 100 µL of phosphate-buffered saline (PBS) intravenously for each inoculation. The administration procedure was the same as with the antibiotic groups.

Before every administration, the animals’ tails were kept in warm water for a few minutes to induce tail vein dilation and then disinfected with 70% alcohol before injecting the needle. The treatment solution and PBS were administered slowly to avoid vascular injury. After the needle was removed, gentle pressure with a sterile compress was applied to stop the bleeding.

### 2.9. Infection Monitoring

All mice were evaluated once daily for the next five days, and any changes in temperature and weight, clinical illness signs, comatose state, or death were noted.

Weak individuals who were unable to sustain themselves were kept hydrated with saline solutions administered subcutaneously and orally.

We assessed the mice using the following scale: 1 = coma; 2 = the animal does not return to the quadrupedal position after being positioned on its back; 3 = the animal returns to the quadrupedal position after 30 s of being positioned on its back; 4 = the animal stands up after 5 s of being positioned on its back, showing minimal motor activity; 5 = normal behaviour [13,28]. Mice assessed as a 1 or 2 on the scale, with no chance of recovery, were humanely euthanised and sampled. Euthanasia was performed using a high dose of anaesthetic.

On the last day, any surviving animals were culled and sampled.

### 2.10. Tissue Sampling and Processing

The animal bodies were processed right after death or euthanasia.

The entire mouse was antiseptised with a 70% ethanol solution and 3% betadine in a sterile laminar airflow work cabinet. Using sterile surgical equipment, a horizontal incision was made at the base of the skull. The skin covering the scalp was pulled over to expose the cranium. The calvaria was detached by cutting the occipital and interparietal bones and then cutting carefully between the frontal and parietal bone junctions with the tip of the scissors to avoid damaging the brain. The calvaria was removed, and the brain was exposed. Using a scalpel, the peduncles and nerves connected with the brain were cut, and the two hemispheres were separated. Half of the brain matter (one hemisphere) was collected into a 2 mL Eppendorf tube for microbiological analysis. The other hemisphere was collected in a histological cassette, submerged in 37% formaldehyde, and sent for histopathological examination to a third-party laboratory.

To reach the spleen, the abdomen was cut open, exposing it on the left side, under the ribcage. Using forceps and scissors, the entire spleen was collected into a 2 mL Eppendorf tube for microbiological analysis.

The samples collected for microbiological analysis were weighed, ground up, and mixed with 1 mL of BHI broth using sterile steel balls and a shaker. From each mix, a 10^−2^ dilution was obtained, and 10 µL of all 4 tubes were inoculated in duplicate on chocolate agar plates supplemented with 10 µg/L vancomycin and 25 µg/L trimethoprim-sulfamethoxazole. The supplement was added to prevent the growth of potential bacterial contaminants, and the neisserial strain was verified to be resistant to the 2 antibiotics at the concentrations mentioned before the experiment. The agar plates were incubated as previously described. After 24 and 48 h, grown colonies were evaluated via MALDI-TOF analysis assay and counted. The results were expressed in CFU/g of brain matter.

The other half of the brain was collected in 37% formaldehyde and sent for histopathological examination to a third-party laboratory.

## 3. Results

### 3.1. Extensive Characterisation of the Neisserial Strain

Our bacterial strain was proven to be susceptible to most antibiotics, as interpreted using the most recent EUCAST guide [19].

The strain was classified as (inhibition diameter shown after the antibiotic and reference diameter for susceptibility shown in brackets) susceptible to the following antibiotics:

Ciprofloxacin 36 mm (≥35 mm), cefotaxim 35 mm (≥34 mm), meropenem 38 mm (≥30 mm), rifampicin 32 mm (≥25 mm), azithromycin 28 mm (≥20), chloramphenicol 26 mm (≥26), and minocycline 28 mm (≥26).

Resistant to trimethoprim-sulfamethoxazole 9 mm (≥30 mm)

During the latex agglutination test, the strain was agglutinated with both serogroup A and serogroup C antibody reagents from the kit. No agglutination has been observed with the serogroup B antibody reagent. The kit’s quality control was performed and revealed no malfunctions (Figure 1). Agglutination for group Y/W-135 was not performed as the manufacturer’s indications do not recommend testing *Neisseria meningitidis* colonies isolated on agar media for Y/W-135 serogroups.

The whole genome sequencing proved that the strain harboured a “capsule null locus” gene instead of any other serogroup capsule-encoding genes (Table 1). It was characterised as a sequence-type (ST) 823 strain from the ST-198 complex [29].

Despite the lack of a capsule, we have identified 56 genes involved in different virulence mechanisms, i.e., 33 genes for adherence, 2 genes for immune modulation, 3 genes for invasion, 11 genes for iron uptake, 1 protease gene, and 6 genes related to stress adaptation (Table 2).

### 3.2. Establishing the Model

The meningococcal meningitis mouse model was established after testing and verifying key variables. In Table 3, the tested variables of each mouse group, as well as survival rate and deaths per day, are presented. The experiments’ succession and results will be presented below in Section 3.2.1 and Section 3.2.2.

#### 3.2.1. Stage 1: Establishing the Required Incubation Time

In the first part of the experiment, we used the protocol described above but with a 24-h incubation time for our BHI broth culture. However, no animals succumbed to the disease.

We decided to increase the incubation time of the culture to 48 h, as this might allow the strain to better express its virulence factors [30]. This time, seven out of eight mice succumbed to the disease. The results can be observed in Figure 2.

#### 3.2.2. Stage 2: Excluding Endotoxic Shock Death, Establishing the Lethal Dose for Half the Inoculated Mice (LD50), and Rising the Survivability Rate through Treatment

All mice from the inactivated inoculum group survived and showed no signs of illness during the six-day period. LD50 was established using a 10^−2^ dilution of the bacterial concentrate (Figure 3).

The survival of mice was significantly different between the 48-h incubated inoculum group and treatment groups. A total of 13 mice succumbed to the disease in the control group, 3 in the ciprofloxacin treatment group, and 4 in the ceftriaxone treatment group (Figure 4). Therefore, the survival rates were 18.75% in the control group, 81.25% in the ciprofloxacin treatment group, and 75% in the ceftriaxone treatment group.

### 3.3. Clinical Parameters and Mortality

At the start of the experiment, the animals weighed an average of 23.7 g and had an average temperature of 36 °C. In all groups where some of the mice succumbed to the disease, the subjects’ clinical parameters deteriorated during the course of the experiment. The temperature and weight seemed to be directly correlated to the health of the animal, decreasing as the animal was going through the onset stages of the disease and gradually increasing in animals that had survived the first 3 days after inoculation. More details regarding the clinical parameters and mortality can be found in the Appendix A.

### 3.4. CFU Count in Tissue Samples

Spleen samples never yielded any bacterial growth throughout the experiment.

In the ceftriaxone treatment group and in the ciprofloxacin treatment group, brain samples never yielded any bacterial growth.

In the summed 48-h incubated control group, mice that either died or were euthanised in the first four days of the experiment presented variable bacterial growth. There was a two-tenfold variation in CFU/g in the brain samples, with the average being 10^3^ CFU/g of brain tissue (Figure 5).

Mice that survived until the 6th day of the experiment never yielded any bacterial growth.

### 3.5. Histologic Examination

Histological analysis of the brain tissue samples with haematoxylin-eosin staining revealed signs of meningitis with leukocytic and haemorrhagic infiltrates (Figure 6 and Figure 7).

Typical meningitis lesions were observed, which were consistent with histological analysis from other mouse models [12], such as abscesses, multiple zones of neutrophilic and lymphocytic infiltrate, and many other signs of inflammation. All of the observed lesions in the PBS-treated mice that succumbed to the disease were also found in the ceftriaxone and ciprofloxacin-treated mice.

## 4. Discussion

We wanted to create an easy-to-replicate meningococcal meningitis mouse model without using expensive mouse strains or hypervirulent *Neisseria meningitidis* strains.

We chose the iron dextran supplement because it was found to be less toxic than other iron compounds, even at higher concentrations [31,32], and it appeared to have a role in the severity of the infection [13,33].

Identical lesions were found in both PBS and antibiotic-treated mouse groups that succumbed to the disease. The histological analysis revealed inflammatory infiltrates and haemorrhages in both the brain parenchyma and the cerebellum, similar to other murine meningitis models [33].

The lack of bacterial growth in all brain tissue samples from the ceftriaxone and ciprofloxacin treatment groups can be explained by the antibiotics still persisting in the tissue samples but failing to save the animal.

The spleen samples never yielded any neisserial growth, despite being one of the main organs that act as antigen filters [34]. This proves that the infection never disseminated throughout the body and was strictly localised in the cerebrum. This further underlines the meningitis aspect of the model rather than septic multi-organ infection.

The model is highly dependent on the bacterial dose inoculum. In the initial phases, while working with the bacterial concentrate (1.5 × 10^10^ CFU/mL), we concluded it was too viscous to administer efficiently with the fine needle syringe. Thus, we used a 10^−1^ dilution, which corresponded to 1.5 × 10^7^ CFU/mouse. This inoculum appeared to be the equivalent of the lethal dose of 90% (LD90) for this strain, as seven out of eight mice died.

A 10^−2^ dilution, which corresponded to 1.5 × 10^6^ CFU/mouse, was obtained and used. In this group, only half of the mice succumbed to the infection (LD50), which was deemed unsatisfactory for the testing of antibiotic compounds for this type of model. However, it is interesting to compare this result with the model developed by Pagliuca et al. [13] using the capsule gene knock-out strain, since they have obtained the LD50 at a value of 10^9^ CFU/mouse. A smaller LD50 could be explained by the fact that our ST-823 strain was naturally capsule-deficient and capable of inducing meningitis in humans and not a knock-out variant of an encapsulated strain. We do not believe this difference is due to the capsule gene knock-out strain lacking virulence factors because the serogroup C strain from which it was obtained was characterised as hypervirulent [13].

After using the 1.5 × 10^7^ CFU/mouse inoculum, six out of eight mice died, and we established the LD80 of the strain by adding in the results from the first group. To lower the number of animals used, the two groups were pooled to create a control group with an identical number of animals as the antibiotic treatment groups.

A peculiarity of the model became apparent in the early stages of development when none of the four inoculated animals succumbed to the infection. The inoculum was prepared using a BHI broth with a 5 mg/L iron dextran culture of our meningococcal strain, which was incubated for 24 h. However, the inoculum optical density and bacterial concentration were the same as in the established protocol. We repeated this 24-h incubation protocol with four more animals, and yet again, all of the animals survived the infection with no signs of illness. The data from these two 24-h incubated inoculum groups were pooled together into a single group and presented in the results section. We obtained the mortality needed in our model only after switching to the 48-h incubation period. This proves, in our opinion, the importance of a longer incubation time for the bacterial inoculum. Most likely, this is attributed to a phenotypic switch in the bacteria, which increased its virulence in the mice.

Another novel part of the model was using the ST-823 capsule-deficient strain. This is a rather rare type of *Neisseria meningitidis* that is not commonly associated with human pathology. However, this was a clinical isolate from a patient who suffered from meningococcal meningitis, and other such cases have been reported [7,35,36,37,38].

Regarding the limitations of the study, we mention the use of only one Neisseria meningitidis strain when developing the model. The LD 50% and LD 80% may vary when using other neisserial strains with different levels of virulence, and concentrations of the bacterial inoculum will have to be adjusted accordingly. Another limitation could be considered the lack of bacterial recovery from mice that succumbed to the disease in the antibiotic treatment groups. Showing that bacteria are still alive in the mice and being able to measure their concentration might prove relevant for certain in-vivo future studies.

The survival rate of treated mice can also be considered a limitation. Both antibiotics did not achieve a 100% survival rate. However, the results of the histological analysis and the clinical outcome are in concordance with human meningococcal meningitis [39,40].

## 5. Conclusions

Our meningococcal meningitis mouse model is not limited to an expensive mouse strain, hypervirulent or encapsulated *Neisseria meningitidis* strain, or antibiotic (as long as the bacterial isolate is susceptible to the antibiotic used and it can penetrate the blood-brain barrier).

The model is inexpensive, simple, and efficient to replicate and implement whenever new antibiotic compounds are to be tested for such a disease.

A novel feature of the model was the use of a BHI broth medium for the bacterial inoculum, a 48-h incubation period of the bacterial inoculum, and the use of an unencapsulated neisserial ST-823 strain from the ST-198 complex.

Another distinguishing feature of the model was that we not only determined the strain’s LD50 and LD80 but also successfully treated and saved mice inoculated with the latter with two clinically used antibiotics (ceftriaxone and ciprofloxacin), demonstrating its utility and flexibility in concordance with other meningococcal murine models used to test new antibacterial compounds [41].

## Figures and Tables

**Figure 1 diagnostics-14-01116-f001:**
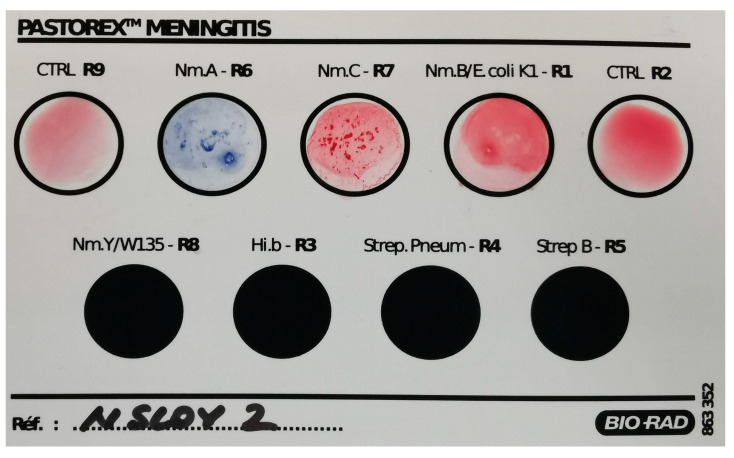
Latex agglutination results of the *Neisseria meningitidis* strain showed non-specific positivity for both serogroups A and C and negative results for serogroup B. Testing for serogroup Y/W135 was not performed as the manufacturer does not recommend using bacterial isolates from agar media. No testing was performed for Hi.b-R3, Strep.Pneum-R4, and Strep B-R5, as these are designated for other bacterial species. (CTRL R9: negative polyvalent control, Nm.A-R6: serogroup A *Neisseria meningitidis*, Nm.C-R7: serogroup C *Neisseria meningitidis*, Nm.B/E.coli K1-R1: serogroup B *Neisseria meningitidis*/*Escherichia coli* K1 antigen, CTRL R2: negative control for serogroup B *Neisseria meningitidis*/*Escherichia coli* K1 antigen, Nm.Y/W135-R8: serogroup Y/serogroup W135 *Neisseria meningitidis*, Hi.b-R3: *Haemophilus influenzae* serotype b, Strep. Pneum-R4: *Streptococcus pneumoniae*, Strep B-R5: group B *Streptococcus)*.

**Figure 2 diagnostics-14-01116-f002:**
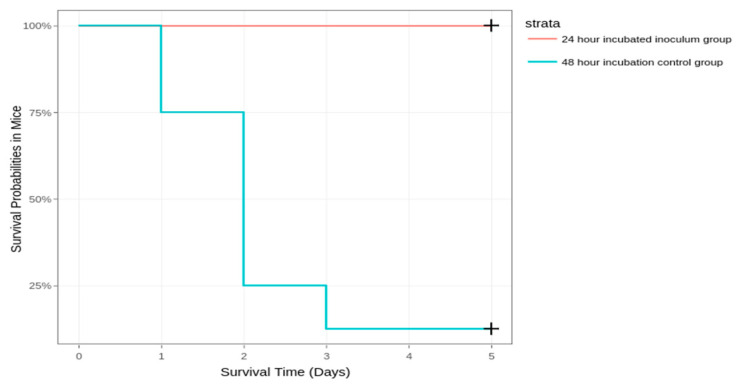
Survival of mice inoculated with the 24-h and 48-h incubated inoculums.

**Figure 3 diagnostics-14-01116-f003:**
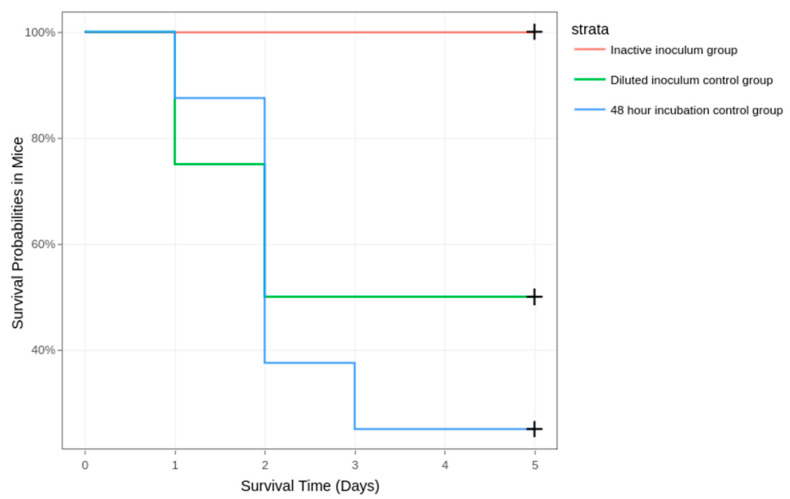
Survival of mice in the inactive inoculum, diluted inoculum (10^−2^), and 48-h incubation group.

**Figure 4 diagnostics-14-01116-f004:**
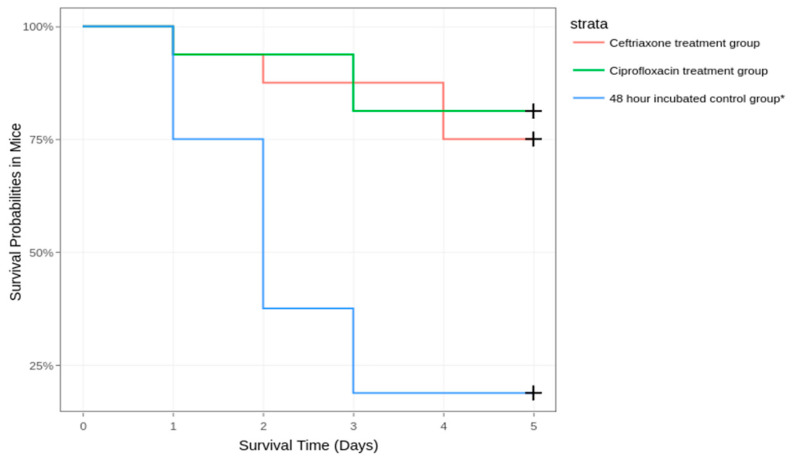
Survival of mice in the two antibiotic treatment groups compared to the 48-h incubated control group* (data pooled from 16 mice from the two 48-h incubated inoculum groups).

**Figure 5 diagnostics-14-01116-f005:**
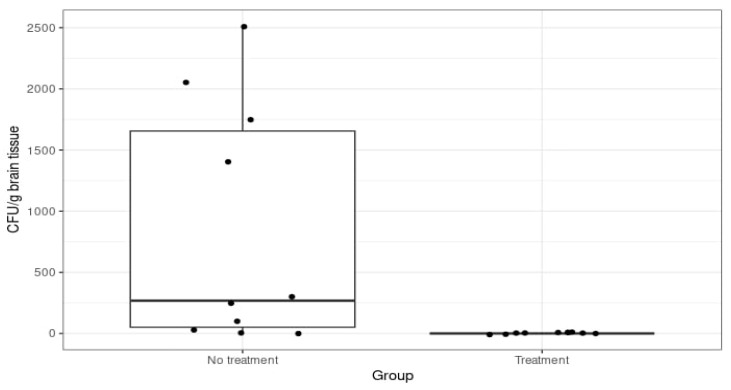
Colony-forming units per gramme of brain tissue in the 48-h incubated control group (No treatment) and the ciprofloxacin and ceftriaxone groups (Treatment).

**Figure 6 diagnostics-14-01116-f006:**
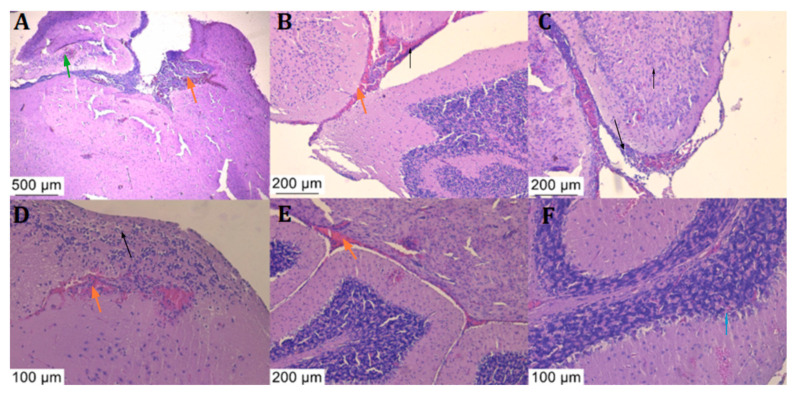
Histological sections of the brains of the animals treated with PBS, stained with haematoxylin and eosin stain: Severe inflammation in the brain parenchyma is characterised by neutrophilic and lymphocytic infiltrate ((**A**–**D**), black arrow) along with haemorrhagic foci both in the brain parenchyma and at its border and at the border with the cerebellum ((**A**–**E**), orange arrow), with perivascular inflammation. Abscess formation is accompanied by inflammation of the vascular endothelium ((**A**), green arrow) and phagocytosis of neutrophils ((**F**), blue arrow).

**Figure 7 diagnostics-14-01116-f007:**
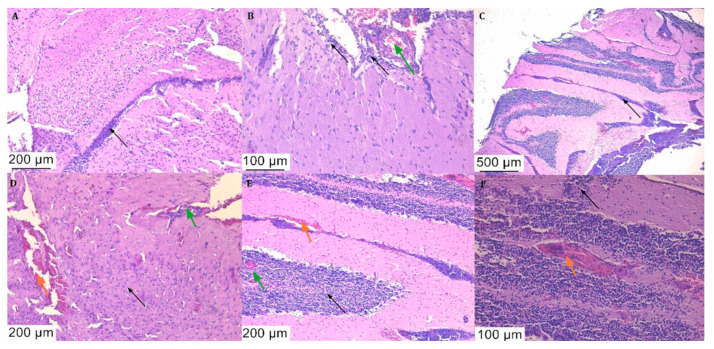
Histological sections of the brains of the ceftriaxone-treated (**A**–**C**) and ciprofloxacin-treated (**D**–**F**) animals that succumbed to the disease, stained with haematoxylin and eosin stain: maintenance of neutrophilic and lymphocytic inflammatory infiltrate ((**A**,**C**), black arrow) or diffuse (black arrow), inflammation of vascular tissues and abscesses delimited by inflammatory reaction ((**B**,**D**), green arrow) together with haemorrhagic foci ((**D**–**F**), orange arrow).

**Table 1 diagnostics-14-01116-t001:** Results of serotyping by MLST based on seven genes [25].

Locus	Identity	Coverage	Alignment Length	Allele Length	Gaps	Allele
abcZ	100	100	433	433	0	abcZ_5
adk	100	100	465	465	0	adk_4
aroE	100	100	490	490	0	aroE_17
fumC	100	100	465	465	0	fumC_15
gdh	100	100	501	501	0	gdh_30
pdhC	100	100	480	480	0	pdhC_7
pgm	100	100	450	450	0	pgm_12

**Table 2 diagnostics-14-01116-t002:** Virulence genes identified using the VFanalyzer tool from the Virulence Factors of Pathogenic Bacteria Data Base (VFDB) [22].

VFclass	Virulence Factor	Gene	Start	Stop	Contig and Orientation
Adherence	Adhesion and penetration protein	app	163	4539	92−
	LOS sialylation	lst	7320	7982	22+
	LOS synthesis	kdtA/waaA	7985	8419	22+
	LOS synthesis	lgtA	3884	5038	51−
	LOS synthesis	lgtB	2334	3860	51−
	LOS synthesis	lgtE	161	1156	27+
	LOS synthesis	lgtF	1212	2741	27+
	LOS synthesis	lgtG	2762	3820	27+
	LOS synthesis	rfaC	88	2229	142−
	LOS synthesis	rfaF	9003	9827	42−
	LOS synthesis	rfaK	230	1156	96+
	Neisseria adhesion A	nadA	1187	3613	96+
	Phosphoethanolamine modification	lptA	10,777	16,284	39−
	Type IV pili	pilC	256	1770	138−
	Type IV pili	pilD	22,475	23,746	19−
	Type IV pili	pilF	244	3081	104−
	Type IV pili	pilG	3078	5291	104−
	Type IV pili	pilH	4889	5977	23−
	Type IV pili	pilI	4050	4889	23−
	Type IV pili	pilJ	3059	3901	23−
	Type IV pili	pilK	3861	4619	101+
	Type IV pili	pilM	2718	3770	43−
	Type IV pili	pilN	14,645	16,279	35+
	Type IV pili	pilO	4323	5459	83+
	Type IV pili	pilP	23,691	24,446	18+
	Type IV pili	pilQ	24,470	25,345	18+
	Type IV pili	pilT	25,422	26,348	18+
	Type IV pili	pilT2	5708	7276	49+
	Type IV pili	pilU	4741	5979	99−
	Type IV pili	pilV	1526	4729	99−
	Type IV pili	pilW	68	1471	99−
	Type IV pili	pilX	5198	6286	9+
	Type IV pili	pilZ	435	959	105+
Immune modulator	Factor H binding protein	fHbp	2729	3397	130−
	Neisserial surface protein A	nspA	2085	2732	130−
Invasion	Class 5 outer membrane protein	opc	1147	2088	130−
	PorA	porA	572	1168	130−
	PorB	porB	11,222	12,337	40+
Iron uptake	ABC transporter	fbpA	12,340	12,939	40+
	ABC transporter	fbpB	12,940	13,587	40+
	ABC transporter	fbpC	13,605	14,150	40+
	Ferric enterobactin transport protein A/ferric-repressed protein B	fetA/frpB	14,169	16,454	40+
	Heme uptake	hpuA	3959	5002	86−
	Heme uptake	hpuB	5462	6592	34+
	Lactoferrin-binding protein	lbpA	2570	3796	86−
	Lactoferrin-binding protein	lbpB	8035	8424	12+
	Ton system	exbB	3049	3810	4+
	Ton system	exbD	94	567	130−
	Ton system	tonB	7607	7957	34+
Protease	IgA protease	iga	458	1627	151−
Stress adaptation	Catalase	katA	375	1370	160+
	Manganese transport system	mntA	8226	9899	66+
	Manganese transport system	mntB	5061	6041	19−
	Manganese transport system	mntC	17,713	18,723	28−
	Methionine sulphoxide reductase	msrA/B (pilB)	4620	5684	101+
	Recombinational repair protein	recN	6412	7254	22+

**Table 3 diagnostics-14-01116-t003:** Mice groups survival throughout the experiment.

Stage	Mouse Group	Mice no.	Inoculum		Deaths						Survival Rate (%)
			Incubation Period	CFU/Mouse	D0	D1	D2	D3	D4	D5	
Stage 1	24-h incubated inoculum group	8	24 h	1.5 × 10^7^	0	0	0	0	0	0	100.00
	48-h incubated inoculum group	8	48 h	1.5 × 10^7^	0	2	4	1	0	0	12.50
Stage 2	Inactivated inoculum group	8	48 h	1.5 × 10^7^	0	0	0	0	0	0	100.00
	Diluted inoculum control group	8	48 h	1.5 × 10^6^	0	2	2	0	0	0	50.00
	48-h incubated control group	8	48 h	1.5 × 10^7^	0	1	4	1	0	0	25.00
	Ceftriaxone treatment group	16	48 h	1.5 × 10^7^	0	1	1	0	2	0	75.00
	Ciprofloxacin treatment group	16	48 h	1.5 × 10^7^	0	1	0	2	0	0	81.25
	Summed 48-h incubated control group *	16	48 h	1.5 × 10^7^	0	3	8	2	0	0	18.75

* The 16 mice from this group were comprised of the pooled results of the 48-h incubated inoculum group and the 48-h incubated control group.

## Data Availability

The raw data supporting the conclusions of this article will be made available by the authors on request.

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
