# Peer review of "Developing a Novel Murine Meningococcal Meningitis Model Using a Capsule-Null Bacterial Strain"

_diagnostics, 2024, doi:10.3390/diagnostics14111116_

Round 1

Reviewer 1 Report

Comments and Suggestions for Authors

In this study, the researchers construct an animal model utilizing a commonly accessible mouse strain and a capsule-deficient strain of Neisseria meningitidis clinically isolated. The meningococcal meningitis mouse model isn't confined to a costly, hyper-virulent, or encapsulated strain of Neisseria meningitidis, nor is it reliant on antibiotic-treated mice. Furthermore, it offers an economical, straightforward, and effective approach to replicate and apply novel antibiotic compounds slated for testing against such a disease. This underscores its utility and adaptability for the scientific community.

Only a few minor errors were detected in the wording of the text: 

Lines 13-14: There is an extra space in the words underlined in yellow

Lines 19, 23 and 27: Subtitles in introduction should be bold

Line 129: mg/L units are used, however later on line 218 ug/l is used. I suggest revising the units of measurement so that they are drafted in the same way

Line 134: Point followed and space to start the new phrase

Line 252: In table 2 the first column is out of alignment and the titles are not read correctly, the same in table 3

In figure 4 there is an asterisk in the text that accompanies the blue line but in the figure caption it is not explained what this sign refers to

Line 311: Discussion in singular

Author Response

Esteemed Reviewer,

Herein we, the Authors, will provide our replies to the commentaries provided in the first round of review for our manuscript. We hope that our response is consistent with the Reviewer’s expectations and that it answers the concerns raised prior.

For ease of reference, we will be including the Reviewer’s notes using quotation marks and italics with our response detailed below the quote.

Respectfully,
The Authors

  • “Lines 13-14: There is an extra space in the words underlined in yellow
  • Lines 19, 23 and 27: Subtitles in introduction should be bold
  • Line 134: Point followed and space to start the new phrase
  • Line 311: Discussion in singular”

We thank the reviewer for the observations. We have modified all the signaled errors.

“Line 129: mg/L units are used, however later on line 218 ug/l is used. I suggest revising the units of measurement so that they are drafted in the same way”

We thank the reviewer for the suggestion. We have changed all measure units to be in the same manner.

“Line 252: In table 2 the first column is out of alignment and the titles are not read correctly, the same in table 3”

We thank the reviewer for the observation. We have changed the dimensions of the written text to better fit the table.

“In figure 4 there is an asterisk in the text that accompanies the blue line but in the figure caption it is not explained what this sign refers to.”

We thank the reviewer for the observation. We have added the needed explanation under the table.

Reviewer 2 Report

Comments and Suggestions for Authors

The work by Viorela-I Caracoti et al is research article in which a novel murine model of meningitis by encapsulated Neisseria meningitis was described.

CD-1 mice were infected by intracisternal inoculum and results were based on healthy status of animals, histological evaluation, bacterial count.

The paper needs to be strongly revised in all parts since abstract is missing of essential information to understand the study, introduction is lacking of appropriate references to support the background reported, methods need to be implemented with clear experimental procedure able for the repetition of experiments, results reports some data but need essential data to support findings, discussion is poor of comparisons with published works and not include discussion of results that authors anticipated.

Please, revise in track changes the work considering:

-        Clearly include the aim of the study in the abstract

-        Check English at line 16.

-        Which is the meaning of incubating bacteria before injection? which conditions? which scientific reasons? Clarify in the abstract, in the introduction.

-        Different prepared inoculums were tested.” Specify the differences of inoculum in the abstract.

-        The bacterial analysis is one of the principal data of the study, and the point was not reported in the abstract.

-        Which is reason for the "cost efficient model" compared to published animal models?

-        Ref 87 refers to meningococcus and not to other causes of meningitis. Move the ref at appropriate place.

-        “Medications” at line 56 refers to therapeutical treatment or preventive methods, or both? It is a useful point to specify to drive the reader for the reason of animal use.

-        Literature with limited application of meningitis models needs to be cited in the text to permit comparison of present background and novelty of the study reported here.

-        Ref 12 is not from Novartis (line 63). Please revise.

-        Report the name of the meningococcal strain at the end of introduction and in materials and methods, to allow repetition of experiment.

-        The use of female animals can be a valid choice but needs to be supported by scientific reason and not by availability of supplier.

-        Info of cage changes can be omitted for the publication.

-        Do you have any national low that refers to EU directive?

-        Clearly report the infection dose of bacteria used to infect animals.

-        All the cryotube stock was inoculated on agar plates? Include the reason of the step (line 110-113).

-        What was inoculate in another agar plate? a colony of the first plate? describe better the method of bacterial culture.

-        Line 114: from plate to a liquid media? Please, clarify the passage.

-        Describe the initial and the second stage of liquid culture of bacteria, underlining the reason of them.

-        Specify not the day of incubation, but the total hour from the inoculum to avoid confusion for the reader.

-        Specify clearly the media used at line 140.

-        Detail how the temperature of animals was evaluated.

-        Include the number of animals used in materials and methods.

-        Detail how the anesthesia was administered to animals.

-        Is it correct that the control includes the meningococcal bacteria treated with warm? if yes, clearly specify in the text: ie. As control, the meningococcal strain used for infection, was inactivated with....

-        Paragraph 2.6 and 2.7 can be included on previous paragraphs (2.3?).

-        Paragraph 2.9 can be included in par. 2.4.

-        Include volume of inoculum and materials and methods used for iv inoculum.

-        How many time the temperature and the weight were evaluated? Report the point in the text.

-        Why were some animals dehydrated if the aim of the study was to evaluate the ability of encapsulated bacteria on clinical parameters and survival as a model for new therapeutical approaches?

-        Describe methods used to collect organs.

-        Why different approach for saline solution administration?

-        Not half brain was reported before line 224. Specify at the beginning that the organ was divided on two parts (dividing hemispheres?). This is essential since the brain in not "homogeneous" organ.

-        Figure 1: describe where serogroup A and C were located, guide the reader to interpret and read the results reported. The use of letter in the figure and in the figure legend can help to interpret position of data.

-        Table 2: explain the acronym of VFDB analsysis at line 252.

-        Table 3: detail the meaning of 24 and 48 h incubated inoculun. Incubation time of the colture with what? which is the scientific significance?

-        Y axis refers to probabilities or observed data? which is the unit used? Can they be expressed in %?

-        Fig 3 and 4 can be reduced in a single fig, considered as controls?

-        What about data (fig. or table) of changes of clinical parameters?

-        What about spleen of animals infected with capsulated bacteria? The point can be included in the discussion.

-        What about histological data? Present data, show figure or table... provide some scientific bases.

-        What about the high dose for the animal health status? Is the reduction of bacterial concentration only dependent on technical issue?

-        Avoid citing "first" or "second" or "final" experiment, but underline the aim of experiments used with all variables you included.

-        Check English line 335.

-        What was performed with the pool of samples between two experiments?

-        Resentence for more clarity lines 345-352.

-        ST-823 is firstly reported at line 353, but it must be cited in previous sections.

-        Discussion for histological analysis and CFU in brain, with comparison of literature, are missing.

-        Include limitations of the study in the discussion.

-        The use of antibiotics was a control of the assay? if you want to support this, cite article that support the thesis and avoid citing articles in which other control were included.

Comments on the Quality of English Language

Extensive editing of English language required.

Author Response

Esteemed Reviewer,

Herein we, the Authors, will provide our replies to the commentaries provided in the first round of review for our manuscript. We hope that our response is consistent with the Reviewer’s expectations and that it answers the concerns raised prior.

For ease of reference, we will be including the Reviewer’s notes using quotation marks and italics with our response detailed below the quote.

Respectfully,
The Authors

Clearly include the aim of the study in the abstract

            We thank the reviewer for the suggestion. We have added the paragraph “In this study we aimed to develop a murine Neisseria meningitidis meningitis model to be used in the study of various antimicrobial compounds”

Check English at line 16.

            We thank the reviewer for pointing out the repetition in our sentence. We have corrected it.

  • Which is the meaning of incubating bacteria before injection? which conditions? which scientific reasons? Clarify in the abstract, in the introduction.
  • “Different prepared inoculums were tested.” Specify the differences of inoculum in the abstract.
  • The bacterial analysis is one of the principal data of the study, and the point was not reported in the abstract.
  • Which is reason for the "cost efficient model" compared to published animal models?”

            We thank the reviewer for the suggestions. We have tried to add the relevant missing information without crossing the word limit of the Abstract.

  • “Ref 87 refers to meningococcus and not to other causes of meningitis. Move the ref at appropriate place.
  • Ref 12 is not from Novartis (line 63). Please revise. “

We thank the reviewer for noticing the misplaced references, we have moved and added them accordingly.

 “ “Medications” at line 56 refers to therapeutical treatment or preventive methods, or both? It is a useful point to specify to drive the reader for the reason of animal use. “

We thank the reviewer for the suggestion. We have added details of the medications with the phrase “aimed at treating confirmed cases of meningococcal meningitis”

“Literature with limited application of meningitis models needs to be cited in the text to permit comparison of present background and novelty of the study reported here. “

We thank the reviewer for the suggestion. Due to that the fact that the phrase ”limited animal models that are dependent on a variety of parameters such” from the abstract also re-appears in the introduction  “meningococcal meningitis mouse models are scarce and rely on different factors” and the abstract has a strict word count, we only left the relevant explanation and citation in the introduction. We cite our Review Article for Murine meningococcal models as a source for an extended comparison of murine meningococcal meningitis models.

“Report the name of the meningococcal strain at the end of introduction and in materials and methods, to allow repetition of experiment. “

We thank the reviewer for the suggestion. We have added the name of the meningococcal strain in this paragraph.

“The use of female animals can be a valid choice but needs to be supported by scientific reason and not by availability of supplier.”

We thank the reviewer for the suggestion. We have removed the reasoning as suggested.

“Info of cage changes can be omitted for the publication.”

We thank the reviewer for the suggestion. We have removed the information.

“Do you have any national low that refers to EU directive?”

We thank the reviewer for the question. Yes, the country has the law 43/2014 regarding protection of animals used for scientific purposes, which was adopted after the 63/2010 EU directive. We have added this information in the text as well.

“Clearly report the infection dose of bacteria used to infect animals.”

We thank the reviewer for this suggestion. The information about the infection dose was written in the following 2.4 paragraph. We have added it in paragraph 2.3 as well to avoid confusion with the paragraph “the inoculum volume was 10 μL this would correspond to 1.5 x 107 CFU/mouse”.

  • “All the cryotube stock was inoculated on agar plates? Include the reason of the step (line 110-113).
  • What was inoculate in another agar plate? a colony of the first plate? describe better the method of bacterial culture.
  • Line 114: from plate to a liquid media? Please, clarify the passage.”

We thank the reviewer for the comments and questions. We have added the inoculated quantities (one 10 μL microbiological loop), the full name of each media (BHI broth media) and an explanation for the steps(in order to obtain a pure culture).

  • “Describe the initial and the second stage of liquid culture of bacteria, underlining the reason of them “

We would like to thank the reviewer for the suggestion. The stages refer to the initial experimental stage where we were still testing out the proper incubation period for the expression of virulence, not the stages of the liquid culture.

  • “Specify not the day of incubation, but the total hour from the inoculum to avoid confusion for the reader.
  • Specify clearly the media used at line 140.”

We thank the reviewer for the suggestions. We have changed the durations from days to hours and specified the name of the media (BHI broth media).

  • “Detail how the temperature of animals was evaluated.
  • Include the number of animals used in materials and methods.”

We thank the reviewer for the suggestions. We have added the inoculated quantities, the name of each media and an explanation for the steps.

“Detail how the anesthesia was administered to animals.”

We thank the reviewer for the suggestions. We have added the information "Mice were anaesthetised intraperitoneally, according to individual weight, with a mixture of ketamine (50 mg/kg) and xylazine (3 mg/kg)."

“Is it correct that the control includes the meningococcal bacteria treated with warm? if yes, clearly specify in the text: ie. As control, the meningococcal strain used for infection, was inactivated with....”

We thank the reviewer for raising up the question of the inactive control. The bacteria was inactivated by heat to show that the deaths of the mice that occurred during the infection model, were not caused by the endotoxin of the dead bacteria, but by its invasiveness and virulence.

“Paragraph 2.6 and 2.7 can be included on previous paragraphs (2.3?)”

We thank the reviewer for the suggestion. However paragraphs 2.6 and 2.7 describe protocols employed when we tested a lower bacterial inoculum and respectively with a shorter incubation time. These represent variations from the protocol described in paragraph 2.3 and we consider they should be kept separate to avoid any misunderstanding regarding the protocol used for inoculum preparation for the proposed model.

“Paragraph 2.9 can be included in par. 2.4.”

We thank the reviewer for the suggestion. We have integrated the information described in paragraph 2.9 into the 2.4 paragraph

“Include volume of inoculum and materials and methods used for iv inoculum. “

We thank the reviewer for the suggestion. We have added the requested information as follows “dose of 50 mg/kg ceftriaxone (corresponding to an average volume of 10 µL/mouse, depending on the animal weight)”, “ IV dose of 25 mg/kg ciprofloxacin (corresponding to an average volume of 200 µL/mouse, depending on the animal weight)” and “Prior to every administration, the animals’ tails were kept in warm water for a few minutes to induce tail vein dilation and then disinfected with 70% alcohol before injecting the needle. The treatment solution and PBS were administered slowly to avoid vascular injury. After the needle was removed, gentle pressure with a sterile compress was applied to stop the bleeding.”

“How many time the temperature and the weight were evaluated? Report the point in the text.”

We thank the reviewer for the observation. We have modified in the paragraph from “daily” to “once per day” to avoid any confusion.

“Why were some animals dehydrated if the aim of the study was to evaluate the ability of encapsulated bacteria on clinical parameters and survival as a model for new therapeutical approaches?”

We thank the reviewer for the question. Due to the acute nature of the infection and clinical observation that mice were transiently weakened so they were unable to hydrate themselves, general supportive measures were implemented. To this end we have observed individual mice which have recovered from this weakened state using only said supportive measures and this is in line with medical clinical practice but abstaining from antibiotic treatment. This was done to properly analyse the mice groups and to avoid deaths caused by dehydration (and not the direct effect of the bacteria) we decided to hydrate the animals.

“Describe methods used to collect organs. “

We thank the reviewer for the suggestion. We have added the paragraphs “a horizontal incision was made at the base of the skull. The skin covering the scalp was pulled over to expose the cranium. The calvaria was detached by cutting the occipital and interparietal bones and then cutting carefully between the frontal and parietal bone junctions with the tip of the scissors to avoid damaging the brain. The calvaria was removed and the brain was exposed. Using a scalpel the peduncles and nerves connected with the brain were cut and the two hemispheres were separated. Half of the brain matter (one hemisphere) was collected into a 2 mL Eppendorf tube for microbiological analysis. The other hemisphere was collected in a histological cassette and submerged in 37% formaldehyde and sent for histopathological examination to a third-party laboratory.” and “To reach the spleen the abdomen was cut open, exposing it on the left side, under the ribcage. Using forceps and scissors, the entire spleen was collected into a 2 mL Eppendorf tube for microbiological analysis”.

“Why different approach for saline solution administration? “

We thank the reviewer for the question. We chose to administer the saline solution intradermally to ensure optimal hydration, similar to a clinical situation, where patients with meningitis are infused with saline solutions, and we also chose to administer it orally because, due to the animals' reduced mobility, it was difficult for them to reach the water source, so it was a kind of aid that we gave to the animals so that they could be hydrated.

“Not half brain was reported before line 224. Specify at the beginning that the organ was divided on two parts (dividing hemispheres?). This is essential since the brain in not "homogeneous" organ.”

We thank the reviewer for the observation. We changed the term into “one hemisphere”.

“ Figure 1: describe where serogroup A and C were located, guide the reader to interpret and read the results reported. The use of letter in the figure and in the figure legend can help to interpret position of data.”

We thank the reviewer for the suggestion. We have added more details regarding the Pastorex kit as well as a legend to guide the reader.

“Table 2: explain the acronym of VFDB analsysis at line 252.”

We thank the reviewer for the suggestion. The acronym VFDB was first explained at line 105. We have added the explanation and complete name.

“Table 3: detail the meaning of 24 and 48 h incubated inoculun. Incubation time of the colture with what? which is the scientific significance?”

We thank the reviewer for the question. The incubation refers to the time that the bacterial strain was grown (incubated) before being used as an inoculum. The scientific significance is that the longer incubated inoculum resulted in more mice succumbing to the disease.

“Y axis refers to probabilities or observed data? which is the unit used? Can they be expressed in %?”

We thank the reviewer for the question. Y axis refers to the observed data. Yes it can be expressed in percentages as well and we have modified all of them.

“Fig 3 and 4 can be reduced in a single fig, considered as controls?”

We thank the reviewer for the question. The two figures contain not only controls, but test groups to verify all parameters. We considered that putting all of the survival curves in the same figure could be hard to follow for the reader.

“What about data (fig. or table) of changes of clinical parameters?”

We thank the reviewer for the question. All the data regarding the clinical parameters can be found in the supplementary material.

“What about spleen of animals infected with capsulated bacteria? The point can be included in the discussion.”

We thank the reviewer for the question. At line 317 in the pdf version of the article we discuss the meaning of the spleen cultures.

“What about histological data? Present data, show figure or table... provide some scientific bases.”

We thank the reviewer for the suggestion. We have added two figures (Figure 6 and Figure 7) with adequate explanations.

“What about the high dose for the animal health status? Is the reduction of bacterial concentration only dependent on technical issue?”

We would like to thank the reviewer for the questions.

Judging from the animal health status in the supplementary material, affected mice in the high dose inoculum and lower dose inoculum groups behaved similarly, with the high dose having a higher mortality than the lower dose.

The reduction of colony forming units / milligram concentration in the brain samples of the treatment groups can be explained by the antibiotic still persisting in the tissue samples, but failing to save the animal.

“Avoid citing "first" or "second" or "final" experiment, but underline the aim of experiments used with all variables you included.”

We thank the reviewer for the suggestion. We have made the changes accordingly.

“Check English line 335.”

We thank the reviewer for the observation. We have modified the sentence to be more precise.

“What was performed with the pool of samples between two experiments?”

We thank the reviewer for the question. The pooled samples refer to the data from two different experimental groups that were added together for the purpose of statistical analysis in a single, virtual group.

“Resentence for more clarity lines 345-352.”

We thank the reviewer for the suggestion. We have rephrased the paragraph for more clarity.

“ST-823 is firstly reported at line 353, but it must be cited in previous sections.”

We thank the reviewer for the suggestion. We have cited the ST-823 in the previous sections where we characterised the strain.

“Discussion for histological analysis and CFU in brain, with comparison of literature, are missing.”

We thank the reviewer for the suggestion. We have added the paragraphs ”Identical lesions were found in both PBS and antibiotic treatment mice groups that succumbed to the disease. The histological analysis revealed inflammatory infiltrate and haemorrhages in both the brain parenchyma and the cerebellum similar to other murine meningitis models [33]” and “The lack of bacterial growth in all brain tissue samples from the ceftriaxone and ciprofloxacin treatment groups can be explained by the antibiotics still persisting in the tissue samples, but failing to save the animal”.

“Include limitations of the study in the discussion.”

We thank the reviewer for the suggestion. We have added the following limitations paragraph: “Regarding the limitations of the study, we mention the use of only one Neisseria meningitidis strain when developing the model. The LD 50% and LD 80% may vary when using other neisserial strains with different levels of virulence and concentrations of the bacterial inoculum will have to be adjusted accordingly. Another limitation could be considered the lack of bacterial recovery from mice that succumbed to the disease from the antibiotic treatment groups. Showing that bacteria is still alive in the mice and being able to measure its concentration might prove relevant for certain in-vivo future studies.

The survival rate of treated mice can also be considered a limitation. Both antibiotics did not obtain a 100% survival rate. However the similarities of the histological analysis of brain samples from PBS and antibiotic treated mice that succumbed to the disease are in concordance with those from other mouse models [33] and the clinical outcome of human meningococcal meningitis [39]”

“The use of antibiotics was a control of the assay? If you want to support this, cite article that support the thesis and avoid citing articles in which other control were included.”

We thank the reviewer for the suggestion. We have cited a similar model “Barman, T.K.; Kumar, M.; Chaira, T.; Gangadharan, R.; Singhal, S.; Rao, M.; Mathur, T.; Bhateja, P.; Pandya, M.; Ramadass, V.; et al. Potential of the Fluoroketolide RBx 14255 against Streptococcus Pneumoniae, Neisseria Meningitidis and Haemophilus Influenzae in an Experimental Murine Meningitis Model. J. Antimicrob. Chemother. 2019, 74, 1962–1970, doi:10.1093/jac/dkz119.”

Round 2

Reviewer 2 Report

Comments and Suggestions for Authors

The authors significantly improved the quality of the manuscript.

Some other issues need to be considered to enhance the scientific soundness:

-        Strongly reduce the extremely detail description of bacteria grown in the abstract (lined 25-28).

-        The speed of centrifugation can be expressed as g since is the best reproducible data.

-        Include in Paragraphs 2.7 the reason for the use of the different protocol described.

-        I don’t understand why in different parts of the manuscript it is underlined the topic of the advantage of the cost-efficient murine model: are actually transgenic mice used for meningitis murine model? For which reason? Is the use of CD-1 instead of more expensive mouse strain the novelty of the study presented?

-        The incubation time of 48 hours of bacteria before inoculum induce more lethality of mice compared to 24 hours incubation: are the bacteria dose, and the respective OD, similar?

-        Good the inclusion of histological data, with figures and relative figure legends. The results text need to be more implemented.

Comments on the Quality of English Language

 Minor editing of English language required

Author Response

Esteemed Reviewer,

Herein, we, the Authors, will reply to the commentaries provided in the second round of review for our manuscript. We hope our response is consistent with the Reviewer’s expectations and answers the prior concerns.

For ease of reference, we will include the Reviewer’s notes in quotation marks and italics, and our response will be detailed below the quote.

Respectfully,

The Authors

“The authors significantly improved the quality of the manuscript.”

We thank the reviewer for the praise!

“Some other issues need to be considered to enhance the scientific soundness:

Strongly reduce the extremely detail description of bacteria grown in the abstract (lined 25-28).”

We thank the reviewer for the suggestion. We have reduced the amount of information in the abstract.

“The speed of centrifugation can be expressed as g since is the best reproducible data.”

We thank the reviewer for the suggestion. We have converted the RPM to G accordingly.

“Include in Paragraphs 2.7 the reason for the use of the different protocol described.”

We thank the reviewer for the suggestion. We have added the explanation of its use: “The purpose of this inoculum was to compare it to the 48-hour inoculum that proved successful in our study”

“I don’t understand why in different parts of the manuscript it is underlined the topic of the advantage of the cost-efficient murine model: are actually transgenic mice used for meningitis murine model? For which reason? Is the use of CD-1 instead of more expensive mouse strain the novelty of the study presented?”

We thank the reviewer for the question. Due to Neisseria meningitidis being a strictly human pathogen, murine model developers need to implement different methods to obtain the infection in mice. Depending on the virulence of strain and the type of mouse used, many models have been described in our murine models review article from ref 12 “Caracoti, V.I.; Caracoti, C.S.; Muntean, A.A.; Coman, C.; Popa, M.I. Proposing a murine meningococcal meningitis animal model based on extensive review of literature. ROAMI 2023,1, 35-41.”. Many of these models implement CD46 transgenic mice to successfully obtain the infection model, which are expensive. CD-1 mice are non-consanguineous and one of the cheapest on the market. It is one of the positive aspects of the model, although other models use CD-1 as well.

“The incubation time of 48 hours of bacteria before inoculum induce more lethality of mice compared to 24 hours incubation: are the bacteria dose, and the respective OD, similar?”

We thank the reviewer for the question. Yes, the bacterial dose and OD of the 24h and 48h incubated inoculums are identical, respectively.

“Good the inclusion of histological data, with figures and relative figure legends. The results text need to be more implemented.”

We thank the reviewer for the praise and the suggestion. We have added more information about the results with the phrase: “Typical meningitis lesions were observed which were consistent with histological analysis from other mouse models [12], such as abscesses, multiple zones of neutrophilic as well as lymphocytic infiltrate and many other signs of inflammation. The observed lesions in the PBS-treated mice that succumbed to the disease were also found in the ceftriaxone and ciprofloxacin-treated mice. “

Round 3

Reviewer 2 Report

Comments and Suggestions for Authors

Revisions strongly improved the Manuscript.